# A Solution for Loneliness in Rural Populations: The Effects of Osekkai Conferences during the COVID-19 Pandemic

**DOI:** 10.3390/ijerph19095054

**Published:** 2022-04-21

**Authors:** Ryuichi Ohta, Koichi Maiguma, Akiko Yata, Chiaki Sano

**Affiliations:** 1Community Care, Unnan City Hospital, 96-1 Iida, Daito-cho, Unnan 699-1221, Japan; 2Department of Law and Economics, Faculty of Law and Literature, Shimane University, 1060 Nishikawatsucho, Matsue 690-8504, Japan; zzkuma@soc.shimane-u.ac.jp; 3Community Nurse Company, 422 Satokata, Kisuki-cho, Unnan 699-1311, Japan; yataakiko0425@gmail.com; 4Department of Community Medicine Management, Faculty of Medicine, Shimane University, 89-1 Enya cho, Izumo 693-8501, Japan; sanochi@med.shimane-u.ac.jp

**Keywords:** Osekkai, social participation, COVID-19, loneliness, community activity, health promotion

## Abstract

Social prescribing is an essential solution to the lack of social connection and interaction and provides a key approach to problems faced by communities during the COVID-19 pandemic. One social prescription used in Japan is the Osekkai conference, which has the potential to increase social participation in rural communities. The revitalization of Osekkai can improve social interactions among people involved in the conference, thereby reducing loneliness. This interventional study was conducted with people who participated in the Osekkai conference. The primary outcomes of the degree of loneliness were compared between 2021 and 2022 using the Japanese version of the three-item UCLA Loneliness Scale. The demographic data and process outcomes of participants were measured using a questionnaire. The participants’ roles in the Osekkai conference increased between the two periods. Loneliness scores tended to decrease during the study period (4.25 to 4.05, *p* = 0.099). In the questionnaire on loneliness, the scores for item 2 decreased significantly during the study period (1.36 to 1.25, *p* = 0.038). In conclusion, this study shows that the continual provision of Osekkai conferences as a social prescription may reduce the degree of loneliness among participants with improved social participation in rural communities. Future studies should investigate comparative interventions to show the effectiveness of social prescription on loneliness in communities.

## 1. Introduction

Social connections are critical for the sustainability of communities in rural areas, a loss of which leads to less social participation. Limited social participation may increase the degree of social isolation and loneliness among people in communities [1,2,3]. Loneliness refers to a condition in which people perceive themselves as socially isolated even when they are among other people, which impinges on their health conditions [2,3]. Social interaction and participation have weakened in these communities because of cultural diversity, the lack of workplaces for the young generation, and privacy issues [4]. Unlike previously, older people in rural communities today do not frequently share their private information and limit their collaboration [5]. The reduced number of people in rural areas and the variety of backgrounds now present have both contributed to decreased interaction [5,6], causing loneliness among older people [7,8]. The COVID-19 pandemic has driven this trend further, as more people in rural areas remain in their homes. The resulting drastic reduction in social connections within communities has the potential to increase the number of people with depression, suicidal intentions, and other mental health problems [9,10,11]. To solve the issue of decreasing social connections and interactions, social prescriptions should be aimed at alleviating community problems.

As a process of addressing aging and isolated societies [12], social prescribing can be essential in approaching community problems and become a solution to losing social connections and interactions. In social prescribing, non-medical interventions related to culture and traditions are proposed to address wider determinants of health, and help patients improve their health behaviors, thereby managing their conditions effectively [12,13]. The variety of social prescriptions are increasing worldwide and these have the potential to establish community sustainability by supporting people’s diverse social lives [14,15].

One social prescription in Japan is the Osekkai conference, offering the possibility of increasing social participation in rural communities. Osekkai is a traditional Japanese behavior in which people do what they think is good for others, which can increase people’s social participation [16]. A previous study showed that participation in Osekkai conferences could be associated with social participation in rural communities [17,18]. Osekkai has been performed as a cultural behavior in Japanese communities. However, aging and multicultural societies may perform Osekkai less frequently, thereby negatively influencing relationships in communities. The revitalization of Osekkai could be a solution for rebuilding social connections in rural communities [16], as rural people feel unjustly suppressed by governmental procedures and social norms due to the COVID-19 pandemic [19,20]. In rural Japanese communities, Osekkai conferences have been held to revitalize Osekkai among rural people during the COVID-19 pandemic since September 2020, leading to the establishment of various new relationships in the communities.

The revitalization of Osekkai can improve the conditions of social interactions among people involved in the conference, which can improve the conditions of loneliness among them. However, no study has clearly shown that participation in Osekkai can improve the conditions of loneliness through an increase in social participation among people involved in Osekkai. Clarification of the effects of social prescribing on the degree of loneliness can drive other rural communities to use social prescribing methods and improve social participation and loneliness. During the COVID-19 pandemic, people in various parts of the world suffered from decreased social interactions, which, in turn, caused multiple health problems [21,22]. This study investigated the effects of Osekkai conferences as social prescriptions on the degree of loneliness among rural people.

## 2. Method

This study utilized an interventional approach to evaluate changes in the degree of loneliness through participation in Osekkai conferences based on a framework from a previous study [16,19,20]. The study was conducted from September 2020 to January 2022. First, we assessed the quality of Osekkai conferences as an intervention using the RE-AIM (Reach, Efficiency/Effectiveness, Adoption, Implementation, and Maintenance) framework [23].

We evaluated changes in the roles at Osekkai conferences among participants to assess how effective the implementation of these conferences were. To examine efficacy and effectiveness, we measured the participants’ change in the degree of loneliness and social participation, in terms of the frequency of meeting with friends or acquaintances, and the number of friends or acquaintances the participant had.

### 2.1. Setting

Unnan City is located in Shimane Prefecture, a rural Japanese prefecture. In 2020, Unnan had a total population of 37,638 (18,145 male and 19,492 female residents), with 39% over 65 years of age; this proportion is expected to reach 50% by 2050 [4]. Unnan City has 30 autonomous community organizations, each with various functions in regard to managing social issues, such as social isolation, accessibility of medical care, and upholding of traditional activities. Each district has at least one autonomous community organization [4].

### 2.2. Osekkai Conference

Both lay and professional rural care providers have begun to hold flexible meetings called Osekkai conferences, which enable rural people to meet and discuss community problems by sharing their knowledge and skills. Individual voluntary activities for communities were created to resolve problems in rural areas, where human resources can be limited, and isolated people often cannot seek help. The conferences began in September 2020 and occur in small groups to avoid the spread of COVID-19 in rural communities [16].

The Unnan City Hall, welfare organizations, and hospitals collaborate and invite various professionals with an in-depth understanding of the city’s social problems and involve participants with a wide variety of skills. The Osekkai conferences involve various community members who present their community’s social issues, such as isolation, unoccupied houses, hoarding disorders, people with physical or mental disabilities who have difficulties with shopping, and older people who have limited human interaction.

The conferences include lay and professional people with a range of professional backgrounds, such as medicine, law, public health, rehabilitation, architecture, community development, and transportation. Participants in the conference collaborate to solve community problems by sharing their experiences and establishing plans for solutions, and, through their dialogue, the Osekkai Plan is established.

The plan is then implemented in rural communities using various resources provided by the communities. The results of the plan’s provisions are then shared at subsequent conferences. Through continued discussions at the conferences, Osekkai plans can be reflected on, and the quality of the Osekkai can be revised [16]. Ongoing discussions can create new and effective collaborations among participants with different resources. The specific activities have been described in a previous open-access article [16].

The resulting increased collaboration can establish various new relationships and activities within communities, which could increase social participation and mitigate loneliness during the COVID-19 pandemic. The framework of the Osekkai conferences is illustrated in Figure 1 [19,20], using an example conference conducted in Kisuki, Unnan City.

### 2.3. Participants

Social media and Unnan City’s local newspapers provided information on the Osekkai conference. People who were interested contacted the office and registered as participants. There were six types of activities related to the Osekkai conference: consulting difficulties in communities to staff at Osekkai conferences, organizing conferences, planning Osekkai, performing Osekkai, supporting Osekkai, and accepting Osekkai. Osekkai conferences were held 16 times during the research. To assess the changes in participants’ social participation and loneliness, an internet-based or letterform questionnaire was provided twice, in February 2021 and February 2022, to all participants at the Osekkai conferences. Both questionnaires had the same contents, and participants who did not use the Internet completed a printed-out questionnaire. The effective response rate was 64.1% (184/287) in 2021 and 72.6% (175/241) in 2022. Of the respondents, 77 completed the questionnaires for both the periods, including the primary outcome. Hence, 77 participants were included in this study (Figure 2).

### 2.4. Measurements

The degree of loneliness was assessed using the Japanese version of the three-item University of California, Los Angeles (UCLA) Loneliness Scale among community-dwelling older adults (score range: 3–9) [24]. The scale consists of three items: Item 1, How often do you feel that you lack companionship? (Scale of 1–3); Item 2, How often do you feel left out? (Scale of 1–3); and Item 3, How often do you feel isolated from others? (Scale of 1–3). We calculated the questionnaire score by summing the score for each item [24]. Additionally, we measured social participation by asking for information regarding the following: the frequency of meeting with friends or acquaintances (more than 4 times/week, 2–3 times/week, once a week, 1–3 times/month, several times/year, no meetings) and the number of friends or acquaintances met within the previous month (no meetings, 1–2 persons, 3–5 persons, 6–9 persons, 10 or more persons) [25]. We also collected data on involvement in activities related to Osekkai conferences (difficulties experienced by communities when consulted Osekkai conference staff, organizing the conference, planning Osekkai, performing Osekkai, supporting Osekkai, and acceptance of Osekkai), age, sex (male or female), residence (Unnan City or elsewhere), social support (have or do not have) [4], receiving regular health checks (yes or no), having chronic diseases (yes or no), smoking (yes or no), alcohol consumption (yes or no), educational background (elementary school, junior high school, high school, and university or higher), and socioeconomic status (rich, relatively rich, not poor, relatively poor, poor).

### 2.5. Analysis

Based on the normality test, we performed a student’s t-test and paired t-test on the parametric data and a Mann–Whitney U test on the nonparametric data. The differences in loneliness scores, frequency of meeting with friends or acquaintances, and roles in the Osekkai conferences were analyzed with a paired t-test and chi-squared test. The variables were categorized binomially: sex (male = 1; female = 0), smoking (yes = 1; no = 0), habitual alcohol drinking (yes = 1; no = 0), educational level (university or more = 1; less than university = 0), social support (having or relative having = 1; not having or relative not having = 0), and socioeconomic status (rich, relatively rich = 1; relatively poor or poor = 0), frequency of meeting with friends or acquaintances (more than 4 times/week or 2–3 times/week or once a week = 1; 1–3 times/month or several times/year, no meeting, or no answer = 0), and the number of friends or acquaintances met within the previous month (more than 10 persons or 6–9 persons = 1, 3–5 persons or 1–2 persons, no meeting, or no answer = 0). Regarding the roles of the Osekkai conference, no response was categorized as no role in each role category. Participants with missing data regarding primary outcomes were excluded from the analysis. Statistical significance was defined as a *p*-value < 0.05, and Cohen’s d was calculated as an effect size.

### 2.6. Ethical Considerations

The participants’ anonymity and confidentiality were ensured throughout the study. The Unnan Public Health Center provided anonymous data, and all of the participants provided written informed consent before participating in the conferences and answering the questionnaire. All procedures in this study were performed in compliance with the Declaration of Helsinki and its subsequent amendments. The Unnan City Hospital Clinical Ethics Committee approved the study protocol (no. 20220021).

## 3. Results

### 3.1. Demographics of the Participants

The average age of the participants was 41.5 years (SD = 16.8), with a male average of 41.8 (SD = 15.4) and a female average of 40.9 (SD = 17.2). Of the participants, 29.9% were male. The percentages of participants with higher education, high socioeconomic status, living with families, living in Unnan City, and high social support were more than 50% (Table 1).

### 3.2. Changes in the Role of Osekkai Conferences during the Study Period

Regarding the roles of the Osekkai conferences, the rates of difficulties in communities, organizing the conference, performing Osekkai, and supporting Osekkai increased between periods, but this was not statistically significant (Table 2).

### 3.3. Changes in the Degree of Loneliness and the Frequency and Numbers of Meeting with Friends or Acquaintances

Cronbach’s alpha for the three questions of the UCLA Loneliness Scale was 0.788 in 2021 and 0.712 in 2022. The two-tailed paired samples *t*-test revealed that the loneliness score was significantly lower in 2022 compared to 2021 (t (76) = 1.67, *p* = 0.099, 95% CI (Confidence interval) [−0.040, 0.456], Cohen’s d = 0.16). In terms of specific items, the two-tailed paired samples t-test revealed a statistically significant decrease in the scores for Item 2 (t (76) = 2.11, *p* = 0.038, 95% CI [0.007, 0.227], Cohen’s d = 0.22). The frequency and number of meetings with friends or acquaintances were persistent during the study period (Table 3).

## 4. Discussion

This study clarified that Osekkai conferences may decrease the degree of loneliness in rural communities through social participation, particularly regarding relationships with friends or acquaintances during the COVID-19 pandemic. Through Osekkai conference participation, considering the increase in the rate of Osekkai performance found in the current research, people might be motivated to engage in social prescription by themselves,. To clarify the effect of social prescription on loneliness and health conditions, future research should involve community-wide investigations in rural communities.

Ongoing provision of Osekkai conferences may motivate participants to efficiently perform Osekkai as a social prescription with the help of other participants, increasing the possibility of spreading social prescriptions as mutual aid within communities. In this study, the participants involved in the Osekkai conference performed Osekkai more than they would have without the conference and did so as a social prescription. In addition, there was an increase in the rates of other Osekkai conference roles during the COVID-19 pandemic. Continuous participation may help the participants understand the effectiveness of social prescription, contributing to their motivation to participate and take on other roles in community social activities [26,27]. Considering the decrease in the Osekkai acceptance rate in general, participants who accepted it might be motivated to perform it with others, increasing the mutual aid process in rural communities.

Continual participation in Osekkai conferences can mitigate the degree of loneliness among people in communities by providing ongoing social participation, improving their health conditions. Community activities, such as recreational activities and exercise programs, can generally be driven by governments [28,29,30]. However, Osekkai conferences can be among the autonomous social activities in which people in communities voluntarily reconsider their relationships and perform social activities that can build their relationships through conference-related activities [31,32]. Although previous research has not clarified this relationship, this study demonstrates the effectiveness of Osekkai conferences in reducing the degree of loneliness among people in communities [19]. The community-oriented issues addressed at the conferences may motivate the participants and generate a volunteer spirit [33,34,35], and Osekkai conferences appear to improve social participation in communities, mitigating loneliness [36,37].

Further investigation should be conducted in the form of community-wide interventions involving a range of stakeholders. At present, the continual provision of Osekkai may temporarily improve the degree of loneliness. A wider range of interventions in communities is essential to demonstrate the effectiveness of social prescribing. Collaboration with various stakeholders is vital to providing social prescriptions widely in communities [38,39]. The interactions of different groups within communities can trigger various problems owing to differences in religion, traditions, and group traits [38,39,40]. These differences may negatively affect collaboration among groups [41]. Engaging in community issue-oriented activities and respecting differences can involve more people in the community and may aid in resolving difficulties [42,43]. Organizing people’s collaboration with respect to their differences can help solve community problems effectively, thus increasing different groups’ social capital. Future research should investigate the quantity and quality of community relationships to understand whether the participants could develop good relationships in their communities.

Furthermore, for the sustainability of social prescribing, activities should be carried out in collaboration with the various stakeholders, such as public organizations, local governments, and community organizations. The involvement of Osekkai conferences in government-driven organizations provides continuity and reliability for citizens, as these are well established and stable. During the COVID-19 pandemic, the sustainability of citizen-driven traditional activities has been disrupted by fear of infection [44,45]. Involving various organizations can provide rural people with numerous resources, such as people, places, and financial support, to continue social prescribing. The continuous provision of social prescribing can be ensured via collaboration with various organizations, as this can lead to the development of sustainable communities and comprehensive care in rural communities [46,47,48].

This study has several limitations. As it was performed in communities located in a rural area of Japan, the study’s setting may be representative only of rural communities in developing and developed countries with a similar lack of medical resources, aging society, and high numbers of isolated older people. Future studies should investigate these constructs in a broader range of communities and other rural settings where people have different cultures that facilitate different behaviors [49]. Another limitation pertains to the sampling methods. Confounding factors were included in this study, and a one-year interval in the measuring schedule could affect the results. Randomization of the sampling process and constant evaluation of the effect of the conferences could further address potential confounding factors. Therefore, future studies should implement randomization to overcome this limitation. Furthermore, the study was a single-arm interventional study with no comparison that affected internal validity. Our intervention was based on the logic models and frameworks established in previous studies [18,23]. The effect of the intervention was assessed based on the endpoint of loneliness and the process outcomes of changes in the roles of Osekkai conferences, which may improve internal and external validity, respectively.

## 5. Conclusions

This study demonstrates that the continual provision of Osekkai conferences as a social prescription can decrease the degree of loneliness among participants by creating social participation in rural communities. Future studies should investigate comparative interventions to demonstrate the effectiveness of social prescription in addressing loneliness in communities.

## Figures and Tables

**Figure 1 ijerph-19-05054-f001:**
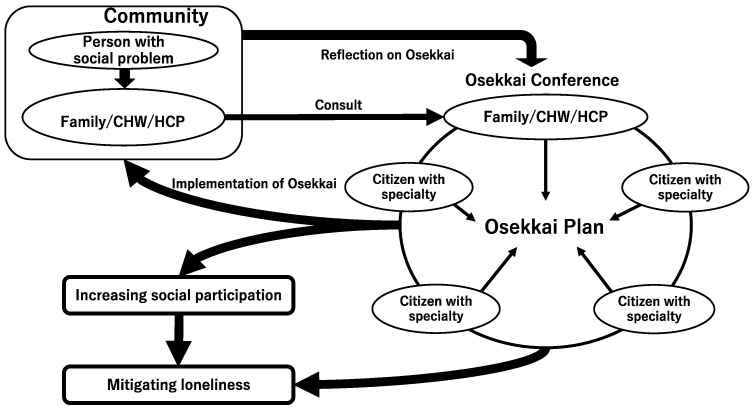
The framework of an Osekkai conference. Notes: CHW, community health worker; HCP, health care practitioner.

**Figure 2 ijerph-19-05054-f002:**
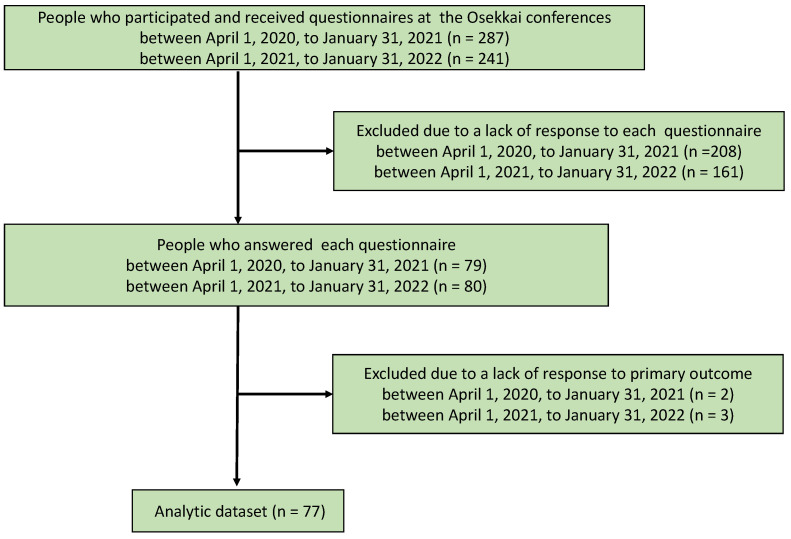
Flow chart of the participants’ selection.

**Table 1 ijerph-19-05054-t001:** Demographics of persons who participated in the Osekkai conference.

Variable	N = 77
Age, mean (SD)	41.5 (16.8)
Male sex (%)	23 (29.9)
High education level (%)	56 (72.7)
High socioeconomic status (%)	51 (66.2)
Consumes alcohol (%)	35 (45.5)
Smoker (%)	3 (3.9)
Regular health check (%)	49 (63.6)
Having chronic diseases (%)	28 (36.4)
Living with family (%)	65 (84.4)
Living location, Unnan (%)	63 (81.8)
High social support (%)	65 (84.4)

**Table 2 ijerph-19-05054-t002:** Change in the role of Osekkai conferences during the study period.

Role in the Conference	2020	2021	*p*-Value
Consulting difficulties in communities	0.43	0.49	0.278
Organizing the conference	0.66	0.73	0.167
Planning Osekkai	0.29	0.30	0.798
Performing Osekkai	0.40	0.45	0.349
Supporting Osekkai	0.35	0.39	0.552
Accepting Osekkai	0.39	0.34	0.397

Notes: Osekkai: Involvement in activities of Osekkai conferences.

**Table 3 ijerph-19-05054-t003:** Change in the degree of loneliness and the frequency and number of instances of meeting with friends or acquaintances.

	Period			
Outcome	2021	2022	*p*-Value	Cohen’s d
Loneliness, mean (SD)	4.25 (1.34)	4.05 (1.21)	0.099	0.16
Item 1, mean (SD)	1.58 (0.55)	1.54 (0.58)	0.605	0.04
Item 2, mean (SD)	1.36 (0.56)	1.25 (0.46)	0.038	0.22
Item 3, mean (SD)	1.31 (0.49)	1.29 (0.51)	0.596	0.04
Meeting frequency	0.39	0.40	>0.99	
Number of meetings	0.55	0.53	>0.99	

Notes: Meeting frequency: The frequency of meeting with friends or acquaintances; Number of meetings: The number of friends or acquaintances met within the previous month.

## Data Availability

The datasets used and/or analyzed during the current study may be obtained from the corresponding author upon reasonable request.

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
