# Peer review of "A Solution for Loneliness in Rural Populations: The Effects of Osekkai Conferences during the COVID-19 Pandemic"

_ijerph, 2022, doi:10.3390/ijerph19095054_

Round 1

Reviewer 1 Report

see review

Author Response

Responses to the reviewers’ editor’s comments

Thank you very much for reviewing our manuscript and providing suggestions for its improvement. We have provided point-by-point responses to the reviewers’ comments; our revisions are indicated in red font here and in the document. We hope that the revised manuscript meets the journal’s requirements and can now be considered for publication.

Reviewer 1

Thank you for inviting me to review this manuscript which presents a rather unique and

interesting study investigating social prescriptions (I do think this is an excellent descriptor) in the form of Osekkai conferences in Japan. Specifically, this research relates to loneliness and the COVID pandemic period. There are a number of issues that preclude its publication at present, BUT the focus of the research is unique and I would strongly urge the authors to address the issues raised here. They are relatively simple but important.

The Introduction lacks a definition of loneliness – given the focus of the research one might expect such a definition.

Response:

Thank you for your valuable feedback. Per your comment, we have added a definition of loneliness in the Background section (Lines 36–37).

Also the authors need to cite the adversity associated with loneliness – the mental and physical health related impact on people, mortality, and social impact - especially in rural areas. Are these impacts different to those in metropolitan areas?

Response:

Thank you for your valuable feedback. Per your comment, we have added more information on rural people’s health conditions because of loneliness during the COVID-19 pandemic (Lines 43–48).

Under method, change in the degree of loneliness is cited as a measure. Quality of friends is important (not just quantity) and while I am aware the authors did not measure this they should bear it in mind for future research.

Response:

Thank you for your valuable feedback. Per your comment, we have added the suggested issue in the Discussion section for future research (Lines 265–268).

Ossekkai conferences (mentioned under method in this study) - How many conferences were involved? Figure 1 is clear and provides a strong visual representation of the method.

Response:

Thank you for your valuable feedback. Per your comment, we have added the number of conferences held during the study (Line 138).

Loneliness and social participation were assessed 12 months apart. Of course it would have been beneficial to have measures in between – say 3 months and 6 months. Was there any particular reason why multiple time point measures were not obtained? This should be mentioned as a limitation because there may have been greater fluctuations in the degree of loneliness in the 12 months between measures.

Response:

Thank you for your valuable feedback. Per your comment, we have added the issues regarding the study duration in the Limitation section (Lines 286–289).

Please provide more details about the questionnaire.

Where did its items come from?

Response:

Thank you for your valuable feedback. Per your comment, we have added an explanation of the questionnaire in the Methods section (Lines 150–155).

How was standardisation of administration obtained? Were there differences in letter and internet administration?

Response:

Thank you for your valuable feedback. Per your comment, we have added an explanation of the administration and provision of the questionnaire in the Methods section (Lines 141–142).

 Was ethics approval obtained and from where? Did the author’s obtain consent from participants?

Response:

Thank you for your valuable feedback. Per your comment, we have revised the ethical considerations, including consent from the participants (Lines 188–193).

What was the percentage response rate for the sample?

Response:

Thank you for your valuable feedback. Per your comment, we have added the statistics to the Methods section (Lines 142–143).

My main concern is with the 3-item UCLA loneliness scale. Although widely used it is a unidimensional measure and as research suggests loneliness is indeed multidimensional.

That is people experience loneliness in different ways and to different levels of intensity. Also 3 items is a minimum number for a measure. Nevertheless, the authors administered this measure. However, the related major issue to this is that they analyzed the loneliness scores separately for each item. The three items together represent the construct of loneliness, therefore the total score of the three items (or their mean score) should be used for the analysis. This must be recomputed.

Response:

Thank you for your valuable feedback. Per your comment, we have added the total score for loneliness in the Results section (Lines 213–219).

Please provide the age range as well as the mean for participants (separately males and females).

Response:

Thank you for your valuable feedback. Per your comment, we have added the average age of each sex in the Results section (Lines 196–197).

Please refer to Tables in text. For example, Table 1 shows ..... or as can be seen in Table 1 ....Similarly, with Table 2.

Response:

Thank you for your valuable feedback. We have cited the tables in the text (Lines 200, 206, and 219).

How was consulting difficulties in communities measured? This is important because of the analyses conducted.

Response:

Thank you for your valuable feedback. Per your comment, we have added an explanation on asking about each role in the conference in the Methods section (Lines 160 - 163).

Please use conventional reporting for t test. Here is an example:

two-tailed paired samples t tests revealed a statistically significant increase in the group’s pre to post self-reported loneliness: t (6) = - 2.82, p = .030, 95% CI [- 3.19, -.23].

Response:

Thank you for your valuable feedback. Per your comment, we have revised the description of the results using the conventional reporting of a paired t-test (Lines 215−218).

Some measure of internal reliability (Cronbach) should be provided (at least for the 3-item loneliness measure). Otherwise the results are questionable.

Response:

Thank you for your valuable feedback. Per your comment, we have added the statistics of Cronbach’s alpha in the Results section (Lines 213−214).

In Table 1 Factor is not an appropriate header. I suggest using variable. Also move the N to where the header Numbers is placed. Delete Numbers and use N as header.

Response:

Thank you for your valuable feedback. Per your comment, we have revised Table 1 comprehensively (Line 201).

Please refer to Table 2 in text. For example, As shown in Table 2 ......

Response:

Thank you for your valuable feedback. We have cited Table 2 in the text (Line 206).

The discussion will require some rewriting depending on the recomputed results.

Response:

Thank you for your valuable feedback. Per your comment, we have revised the Discussion section comprehensively based on the revision of the previous section (Line 223).

I like this manuscript because it highlights a highly unique approach which I believe has cultural importance. Should the authors require clarification I am more than pleased to assist.

Thank you for your valuable time.

Reviewer 2 Report

The authors adopted a social intervention method, which is named “Osekkai conference”, to improve social interactions among people in the rural areas in Japan. To test the effectiveness of this method, an Osekkai intervention was conducted from September 2020 to January 2021. Loneliness decreased during the study period (p = 0.099)

I think the topic is important and the study itself is interesting, but there are still concerns, which are listed below:

  1. One important thing about scientific communication is that authors must provide all the detailed information for future researchers to replicate. However, based on the information the authors have provided in the manuscript, I don't think readers are able to reproduce the same procedure. I believe there must be some “script” for the Osekkai conference, like how the host initiated the conference, what topics exactly were the participants talking about, what plan have the participants made after the conference. The authors could attach the detailed script in the supplementary document.
  2. The authors did mention that they use the t-test and Mann Whitney U test. But in the result section, the authors only mentioned p value without specifying the statistical method they used. Please add information such as t (xx) = xx, p = .09.
  3. One major concern is that the statistical results did not reach significance for the index of loneliness (p = .099). I understand that we can’t totally rely on the p value. However, nowadays in social science, if you don’t want to rely on p value, you need to provide indices of effect size (Cohen’s d, eta squared, etc) for readers to evaluate the effect size.
  4. Following the previous question, I believe that the effect size on loneliness, based on the p value, and the sample size of 77, should not be large. In that case, I would suggest the authors to soften the tone across the whole manuscript. For example, on P6, Line 4, “Loneliness score decreased during the study period” should be rephrased as “there was a trend of decreased loneliness score….”. Also, the first line of Discussion, “this study clarified that Osekkai conference can decrease the degree of loneliness” should be re-written. One should deal with statistical insignificant results more carefully.
  5. Normally, when we measure one construct with multiple items, we provide the index of reliability, which is Cronbach’s alpha. Therefore, I suggest the authors to provide the Cronbach’s alpha value for their loneliness measurement. Furthermore, it’s a bit weird to provide information about individual item as shown in table 3.
  6. Although loneliness did not reach significance, it was at least marginal significance. However, for “change in the role” as shown in Table 2, all the indices were far from significance. Therefore, I don’t think the data were strong enough to for the authors to draw a conclusion as shown in the last paragraph on p5, “Regarding the roles of the Osekkai conferences, the rates of consulting difficulties in communities, organizing the conference, performing Osekkai, and supporting Osekkai increased between periods. However, the role of accepting Osekkai decreased.” Basically no significant difference was found in changes in the role before and after the Osekkai conference.
  7. The conclusion, the second line, “social prescription can improve the degree of loneliness” The degree of loneliness can only be increased or decreased, but cannot be “improved”.
  8. In Table 3, there are some p values that equal 1. The p value indicates a probability, so it cannot be 0 or 1. If it is really large, the author should change it into >.99

Author Response

Responses to the reviewers’ editor’s comments

Thank you very much for reviewing our manuscript and providing suggestions for its improvement. We have provided point-by-point responses to the reviewers’ comments; our revisions are indicated in red font here and in the document. We hope that the revised manuscript meets the journal’s requirements and can now be considered for publication.

Reviewer 2

The authors adopted a social intervention method, which is named “Osekkai conference”, to improve social interactions among people in the rural areas in Japan. To test the effectiveness of this method, an Osekkai intervention was conducted from September 2020 to January 2021. Loneliness decreased during the study period (p = 0.099)

I think the topic is important and the study itself is interesting, but there are still concerns, which are listed below:

  1. One important thing about scientific communication is that authors must provide all the detailed information for future researchers to replicate. However, based on the information the authors have provided in the manuscript, I don’t think readers are able to reproduce the same procedure. I believe there must be some “script” for the Osekkai conference, like how the host initiated the conference, what topics exactly were the participants talking about, what plan have the participants made after the conference. The authors could attach the detailed script in the supplementary document.

Response:

Thank you for your valuable feedback. Per your comment, we have revised the explanation of the Osekkai conferences comprehensively by adding specific explanations and adding a reference, including the activities established by the Osekkai conferences (Lines 101–128).

  1. The authors did mention that they use the t-test and Mann Whitney U test. But in the result section, the authors only mentioned p value without specifying the statistical method they used. Please add information such as t (xx) = xx, p = .09.

Response:

Thank you for your valuable feedback. Per your comment, we have revised the description of the results by using the conventional reporting for a paired t-test (Lines 215−218).

  1. One major concern is that the statistical results did not reach significance for the index of loneliness (p = .099). I understand that we can’t totally rely on the p value. However, nowadays in social science, if you don’t want to rely on p value, you need to provide indices of effect size (Cohen’s d, eta squared, etc) for readers to evaluate the effect size.

Response:

Thank you for your valuable feedback. Per your comment, we have revised the description of the results by adding Cohen’s d to the results of the loneliness score (Lines 216−218).

  1. Following the previous question, I believe that the effect size on loneliness, based on the p value, and the sample size of 77, should not be large. In that case, I would suggest the authors to soften the tone across the whole manuscript. For example, on P6, Line 4, “Loneliness score decreased during the study period” should be rephrased as “there was a trend of decreased loneliness score….”. Also, the first line of Discussion, “this study clarified that Osekkai conference can decrease the degree of loneliness” should be re-written. One should deal with statistical insignificant results more carefully.

Response:

Thank you for your valuable feedback. Per your comment, we have revised the description to tone down the interpretation of the results.

  1. Normally, when we measure one construct with multiple items, we provide the index of reliability, which is Cronbach’s alpha. Therefore, I suggest the authors to provide the Cronbach’s alpha value for their loneliness measurement. Furthermore, it’s a bit weird to provide information about individual item as shown in table 3.

Response:

Thank you for your valuable feedback. Per your comment, we have added the statistics for Cronbach’s alpha in the Results section (Lines 213−214).

  1. Although loneliness did not reach significance, it was at least marginal significance. However, for “change in the role” as shown in Table 2, all the indices were far from significance. Therefore, I don’t think the data were strong enough to for the authors to draw a conclusion as shown in the last paragraph on p5, “Regarding the roles of the Osekkai conferences, the rates of consulting difficulties in communities, organizing the conference, performing Osekkai, and supporting Osekkai increased between periods. However, the role of accepting Osekkai decreased.” Basically no significant difference was found in changes in the role before and after the Osekkai conference.

Response:

Thank you for your valuable feedback. Per your comment, we have revised the description by adding the description for the results not being statistically significant.

  1. The conclusion, the second line, “social prescription can improve the degree of loneliness” The degree of loneliness can only be increased or decreased, but cannot be “improved”.

Response:

Thank you for your valuable feedback. Per your comment, we have revised the sentence (Line 298).

  1. In Table 3, there are some p values that equal 1. The p value indicates a probability, so it cannot be 0 or 1. If it is really large, the author should change it into >.99

Response:

Thank you for your valuable feedback. Per your comment, we have revised the description of the p-value to >0.99 (Line 220).

Round 2

Reviewer 2 Report

The quality of the paper has been significantly improved. My only suggestion is that in the descriptions of Cohen'd, the statistic values should be listed as t (xx) = xx, p = xx, Cohen's d = ... It should be a "=" instead of a "," after Cohen's d. 

Author Response

Dear reviewer, I am grateful that you have found our previous revisions to your satisfaction. Your suggestions and feedback have significantly helped us strengthen our manuscript. Below is a response to your query regarding the reporting of statistics.

The quality of the paper has been significantly improved. My only suggestion is that in the descriptions of Cohen'd, the statistic values should be listed as t (xx) = xx, p = xx, Cohen's d = ... It should be a "=" instead of a "," after Cohen's d. 

Response:

Thank you for your keen attention and valuable feedback. Per your comment, we have revised the reporting of statistical values in the Result section, specifically the description of Cohen’s d (Lines 213–219).